# DynVideo-E: Harnessing Dynamic NeRF for Large-Scale Motion- and View-Change Human-Centric Video Editing

## Abstract

Despite remarkable research advances in diffusion-based video editing, existing methods are limited to short-length videos due to the contradiction between long-range consistency and frame-wise editing. Recent approaches attempt to tackle this challenge by introducing video-2D representations to degrade video editing to image editing. However, they encounter significant difficulties in handling large-scale motion- and view-change videos especially for human-centric videos. This motivates us to introduce the dynamic Neural Radiance Fields (NeRF) as the human-centric video representation to ease the video editing problem to a 3D space editing task. As such, editing can be performed in the 3D spaces and propagated to the entire video via the deformation field. To provide finer and direct controllable editing, we propose the image-based 3D space editing pipeline with a set of effective designs. These include multi-view multi-pose Score Distillation Sampling (SDS) from both 2D personalized diffusion priors and 3D diffusion priors, reconstruction losses on the reference image, text-guided local parts super-resolution, and style transfer for 3D background space. Extensive experiments demonstrate that our method, dubbed as DynVideo-E, significantly outperforms SOTA approaches on two challenging datasets by a large margin of $50\% \sim 95\%$ in terms of human preference. Compelling video comparisons are provided in the anonymous project page `https://dynvideo-e.github.io`. Our code and data will be released to the community.

## 1 Introduction

The remarkable success of powerful image diffusion models has sparked considerable interests in extending them to support video editing. Despite promising, it presents significant challenges in maintaining high temporal consistency. To tackle this problem, existing diffusion-based video editing approaches have evolved to extract and incorporate various correspondences from source video into the frame-wise editing process, including attention map (Qi et al., 2023; Liu et al., 2023c), spatial map (Yang et al., 2023; Zhao et al., 2023), optical flow and nn-fields (Geyer et al., 2023). While these works have demonstrated enhanced temporal consistency of editing results, the inherent contradiction between long-range consistency and frame-wise editing limits these methods to short videos with small motion and viewpoint changes.

Another line of research seeks to introduce intermediate video-2D representations to degrade video editing to image editing, such as decomposing videos using the layered neural atlas (Kasten et al., 2021) and mapping spatial-temporal contents to 2D UV maps. As such, editing can be performed on a single frame (Huang et al., 2023; Lee et al., 2023) or on the atlas itself (Chai et al., 2023; Couairon et al., 2023; Bar-Tal et al., 2022), with the edited results consistently propagating to other frames. More recently, CoDeF (Ouyang et al., 2023) proposes the 2D hash-based canonical image coupled with a 3D deformation field to further improve the video representative capability. However, these approaches are 2D representations of video contents, and thus they encounter significant difficulties in representing and editing videos with large-scale motion and viewpoint changes.

This motivates us to introduce the video-3D representation for large-scale motion- and view-change video editing. Recent advances in dynamic NeRF (Liu et al., 2023a; Weng et al., 2022; Jiang et al.,

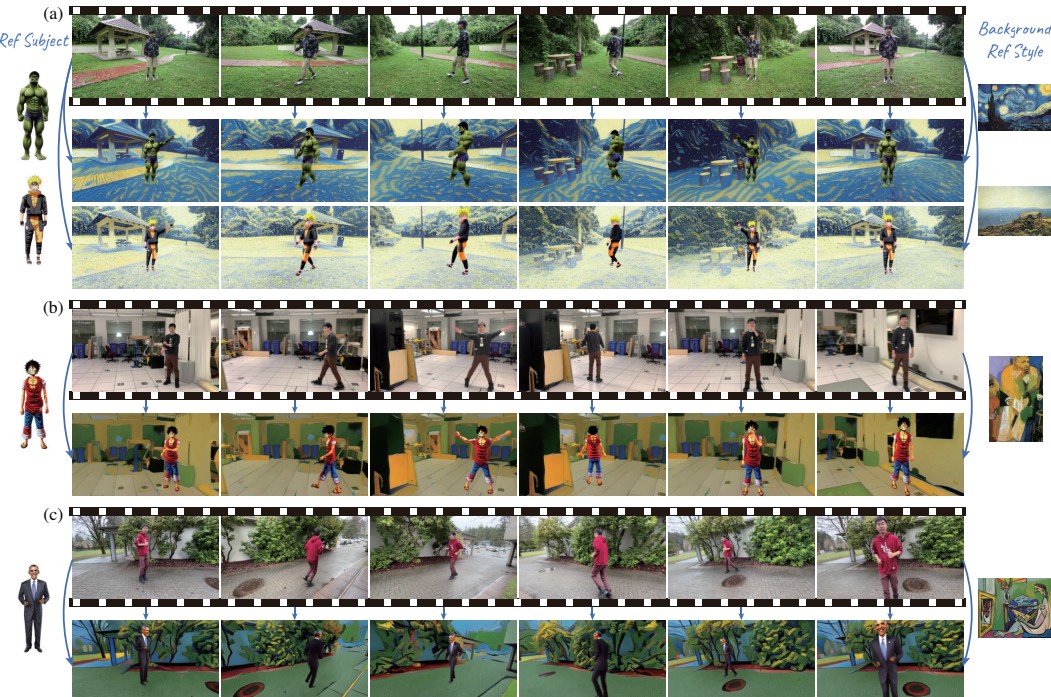

Figure 1: Our DynVideo-E enables consistent editing of large-scale motion- and view-change human-centric videos, given reference subject image and background style image. Details such as identity and clothes textures are well preserved with very high temporal consistency (a-c).

2022) show that the 3D human canonical space coupled with the human pose guided deformation field can effectively reconstruct single human-centric videos with large deformations and viewpoints changes. Therefore, in this paper, we focus on human-centric video editing and propose DynVideo-E that for the first time introduces dynamic NeRF as the video representation to ease the video editing task into a 3D space editing problem. Such a video-NeRF representation effectively aggregates the large-scale motion- and view- change video information into a 3D background space and a 3D canonical human space through the reconstructed human pose guided deformation fields, and the edited contents can be accurately propagated to the entire video via the deformation fields.

To provide finer and direct controllable editing, we propose the image-based 3D space editing pipeline with a set of effective strategies customized for our video-NeRF representation. These include 1) reconstruction losses on the reference image under reference human pose and camera viewpoint to inject personalized contents from the reference image to the edited 3D human space. 2) To improve the 3D consistency and animatibility of the edited human space, we design a multi-view multi-pose Score Distillation Sampling (SDS) from both 2D personalized diffusion priors and 3D diffusion priors, as well as a set of training strategies under various human pose and camera pose configurations. 3) To improve the resolution and geometric details of 3D human space, we utilize the text-guided local parts zoom-in super-resolution with 7 semantic body regions augmented with view conditions. 4) We employ a style transfer module to transfer the reference style to our 3D background model. After training, our video-NeRF model can render highly consistent videos along source video viewpoints by propagating the edited contents through the deformation field.

We extensively evaluate our DynVideo-E on HOSNeRF (Liu et al., 2023a) and NeuMan (Jiang et al., 2022) dataset with 24 editing prompts on 11 challenging dynamic human-centric videos. As shown in Fig. 1, our DynVideo-E generates photorealistic video editing results with very high temporal consistency, and significantly outperforms SOTA approaches by a large margin of $50\% \sim 95\%$ in terms of human preference.

To summarize, the major contributions of our paper are:

- We present a novel framework of DynVideo-E that for the first time introduces the dynamic NeRF as the innovative video representation for large-scale motion- and view-change human-centric video editing.

- We propose a set of effective designs and training strategies customized for the image-based 3D canoincal and background space editings in our video-NeRF model.

- Extensive experiments show that DynVideo-E significantly outperforms SOTA approaches on two challenging datasets by a large margin of $50\% \sim 95\%$ in terms of human preference.

## 2 RELATED WORK

### 2.1 DIFFUSION-BASED VIDEO EDITING

Thanks to the power of diffusion models, prior works have extended their support to video editing (Wu et al., 2022) and generation (Zhang et al., 2023a). Pioneer Tune-A-Video (Wu et al., 2022) inflates the image diffusion with cross-frame attention and fine-tunes the source video, aiming to implicitly learn the source motion and transfer it to the target video. Although Tune-A-Video demonstrates versatility across different video editing applications, it exhibits inferior temporal consistency. Subsequent works extract various correspondences from the source video and employ them to improve temporal consistency. FateZero (Qi et al., 2023) and Video-P2P (Liu et al., 2023c) extract the cross- and self-attention from the source video to control spatial layout. Rerender-A-Video (Yang et al., 2023), ControlVideo (Zhao et al., 2023), and TokenFlow (Geyer et al., 2023) extract and align optical flow, spatial maps, and nn-fields from the source video, resulting in improved consistency. Although these works have shown promising results, they are typically used in short-form video editing scenarios with small-scale motions and view changes.

Another significant line of video editing work relies on a powerful video representation, namely, the layered neural atlas (Kasten et al., 2021), as an intermediate editing representation. The layered neural atlas factorizes the input video using a layered presentation and maps the subject/background of all frames using 2D UV maps. Once the layered neural atlas is learned, editing can occur either on keyframes (Huang et al., 2023; Lee et al., 2023) or on the atlas itself (Chai et al., 2023; Couairon et al., 2023; Bar-Tal et al., 2022), and the editing results consistently propagate to other frames. CoDeF (Ouyang et al., 2023) incorporates the 3D deformation field with the 2D hash-based canonical image to further improve the video representative capability. However, both the layered neural atlas (Kasten et al., 2021) and canonoical image (Ouyang et al., 2023) are pseudo-3D representations of video contents, and they encounter difficulties in reconstructing videos with 3D view changes.

### 2.2 DYNAMIC NERFS

Remarkable progress has been made in the field of novel view synthesis since the introduction of Neural Radiance Fields (NeRF) (Mildenhall et al., 2021). Subsequent studies have extended it to reconstruct dynamic NeRFs from monocular videos by either learning a deformation field that maps dynamic space to canonical field (Pumarola et al., 2021; Park et al., 2021a;b; Tretschk et al., 2021) or building 4D spatio-temporal radiance fields (Xian et al., 2021; Li et al., 2021; Gao et al., 2021). Other studies have introduced voxel grids (Liu et al., 2022; Fang et al., 2022; Song et al., 2022) or planar representations (Fridovich-Keil et al., 2023; Cao & Johnson, 2023) to improve the training efficiency of dynamic NeRFs. While these approaches have shown promising results, they are limited to short videos with simple deformations. Another series of work focus on human-centric modelling and leverage estimated human poses priors (Peng et al., 2021; Weng et al., 2022) to reconstruct dynamic humans with complex motions. Recently, Neuman (Jiang et al., 2022) reconstructs the dynamic human NeRF together with static scene NeRF to model human-centric scenes. HOSNeRF (Liu et al., 2023a) further proposes to represent complex human-object-scene with state-conditional dynamic human model and unbounded background model, achieving 360° free-viewpoint renderings from single videos. In contrast, we aim to introduce the dynamic NeRF as the video-NeRF representation for human-centric video editings.

### 2.3 NERF-BASED EDITING AND GENERATION

Since the introduction of diffusion models, text-guided 3D NeRF editing and generation has evolved from CLIP-based (Wang et al., 2023; Hong et al., 2022) to 2D diffusion-based (Zhuang et al., 2023; Mikaeili et al., 2023; Sella et al., 2023; Li et al., 2023; Kolotouros et al., 2023). In contrast, SINE (Bao et al., 2023) supports editing a local region of static NeRF from a single view

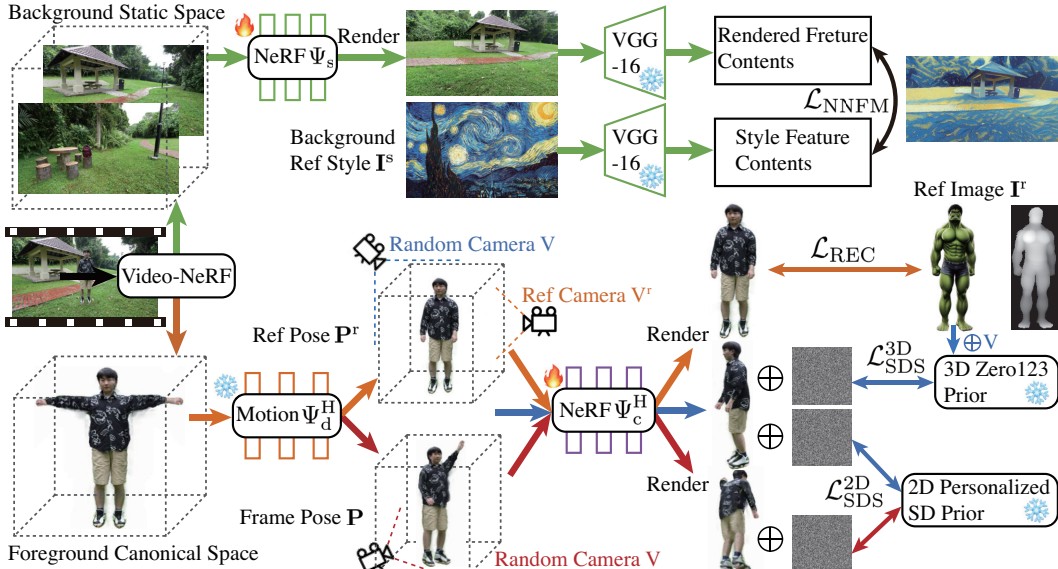

Figure 2: **Overview of DynVideo-E.** (1) Our **video-NeRF model** represents the input video as the 3D foreground canonical human space coupled with the deformation field and the 3D background static space. (2) **Orange flowchart:** Given the reference subject image, we edit the animatable canonical human space under multi-view multi-pose configurations by leveraging reconstruction losses, 2D personalized diffusion priors, 3D diffusion priors, and local parts super-resolution. (3) **Green flowchart:** A style transfer loss in feature spaces is utilized to transfer the reference style to our 3D background model. (4) **Edited videos** can be accordingly rendered by volume rendering in the edited video-NeRF model under source video camera poses.

by delivering edited contents to multi-views through pretrained NeRF priors. More recently, Control4D (Shao et al., 2023) and Dyn-E (Zhang et al., 2023b) stands out as the pioneering work to edit dynamic NeRFs. However, Control4D (Shao et al., 2023) is limited to human-only videos with small motions and short video length, while Dyn-E (Zhang et al., 2023b) only supports editing the local appearance of dynamic NeRFs trained on short-length videos.

## 3 METHOD

### 3.1 VIDEO-NERF MODEL

**Motivation.** Given single videos with large viewpoint changes, intricate scene contents, and complex human motions, we seek to represent such videos using dynamic NeRFs to ease the video editing problem to a static 3D editing task. HOSNeRF (Liu et al., 2023a) has been recently proposed to reconstruct the dynamic neural radiance field for dynamic human-environment interactions from a single monocular in-the-wild video and achieves new SOTA performances. It proposes the state-conditional dynamic human-object model and background model to separately represent the dynamic human and static background. Since we mainly focus on dynamic human-centric videos, we simplify HOSNeRF (Liu et al., 2023a) to HSNeRF by removing the object state designs, and we introduce the HSNeRF as our video-NeRF model for large-scale motion- and view-change video editing, which consists of a dynamic human model $\Psi^{\mathrm{H}}$ and a static scene model $\Psi^{\mathrm{S}}$.

**Dynamic Human Model** $\Psi^{\mathrm{H}}$ aggregates the dynamic information across all video frames into a 3D canonical human space $\Psi_{\mathrm{c}}^{\mathrm{H}}$ that maps 3D points to color $\mathbf{c}$ and density $d$, and a human pose-guided deformation field $\Psi_{\mathrm{d}}^{\mathrm{H}}$ that maps deformed points $\mathbf{x}_{\mathrm{d}}^{i}$ from the deformed space at frame $i$ to canonical points $\mathbf{x}_{\mathrm{c}}^{i}$ in the canonical space ($i$ omitted for simplicity).

$$\Psi_{\mathrm{c}}^{\mathrm{H}}\left(\gamma\left(\mathbf{x}_{\mathrm{c}}\right)\right) \longmapsto (\mathbf{c}, d), \quad \Psi_{\mathrm{d}}^{\mathrm{H}}\left(\mathbf{x}_{\mathrm{d}}, \mathcal{J}, \mathcal{R}\right) \longmapsto \left(\mathbf{x}_{\mathrm{c}}\right), \qquad (1)$$

where $\gamma\left(\mathbf{x}\right)$ is the standard positional encoding function, and $\mathcal{J} = \{\mathbf{J}_{i}\}$ and $\mathcal{R} = \{\boldsymbol{\omega}_{i}\}$ are 3D human joints and local joint axis-angle rotations, respectively.

Following HOSNeRF (Liu et al., 2023a) and HumanNeRF (Weng et al., 2022), we decompose the deformation field $\Psi_{\mathrm{d}}^{\mathrm{H}}$ into a coarse human skeleton-driven deformation $\Psi_{\mathrm{d}}^{\mathrm{H,coarse}}$ and a fine non-rigid deformation conditioned on human poses $\Psi_{\mathrm{d}}^{\mathrm{H,fine}}$:

$$\mathbf{x}_{\mathrm{c}}' = \Psi_{\mathrm{d}}^{\mathrm{H,coarse}}\left(\mathbf{x}_{\mathrm{d}},\,\mathcal{J},\,\mathcal{R}\right), \quad \mathbf{x}_{\mathrm{c}} = \mathbf{x}_{\mathrm{c}}' + \Psi_{\mathrm{d}}^{\mathrm{H,fine}}\left(\mathbf{x}_{\mathrm{c}}',\,\mathcal{R}\right). \tag{2}$$

**Static Scene Model** $\Psi^{\mathrm{S}}$ aggregates intricate static scene contents into a Mip-NeRF 360 (Barron et al., 2022) space that maps contracted Gaussian parameters $(\hat{\boldsymbol{\mu}}, \hat{\boldsymbol{\Sigma}})$ to color $\mathbf{c}$ and density $d$.

$$\Psi_{\mathrm{s}}\left(\hat{\gamma}\left(\hat{\boldsymbol{\mu}},\,\hat{\boldsymbol{\Sigma}}\right)\right) \longmapsto (\mathbf{c},\,\sigma), \tag{3}$$

where $\hat{\gamma}$ is the integrated positional encoding (IPE) (Barron et al., 2022):

$$\hat{\gamma}(\hat{\boldsymbol{\mu}}, \hat{\boldsymbol{\Sigma}}) = \left\{ \begin{bmatrix} \sin(2^\ell \hat{\boldsymbol{\mu}}) \exp\left(-2^{2\ell-1} \operatorname{diag}\left(\hat{\boldsymbol{\Sigma}}\right)\right) \\ \cos(2^\ell \hat{\boldsymbol{\mu}}) \exp\left(-2^{2\ell-1} \operatorname{diag}\left(\hat{\boldsymbol{\Sigma}}\right)\right) \end{bmatrix} \right\}_{\ell=0}^{L-1}. \tag{4}$$

To obtain the contracted Gaussian parameters, we first split the casted rays into a set of intervals $T_i = [t_i, t_{i+1})$ and compute their corresponding conical frustums' mean and covariance as $(\boldsymbol{\mu}, \boldsymbol{\Sigma}) = \mathbf{r}(T_i)$ (Barron et al., 2022). Then we adopt the contraction function $f(\mathbf{x})$ proposed in Mip-NeRF 360 (Barron et al., 2022) to distribute distant points proportionally to disparity, and parameterize the Gaussian parameters for unbounded scenes as follows,

$$f(\mathbf{x}) = \begin{cases} \mathbf{x} & \|\mathbf{x}\| \le 1 \\ \left(2 - \frac{1}{\|\mathbf{x}\|}\right)\left(\frac{\mathbf{x}}{\|\mathbf{x}\|}\right) & \|\mathbf{x}\| > 1 \end{cases}, \tag{5}$$

and $f(\mathbf{x})$ is applied to $(\boldsymbol{\mu}, \boldsymbol{\Sigma})$ to obtain contracted Gaussian parameters:

$$\left(\hat{\boldsymbol{\mu}},\,\hat{\boldsymbol{\Sigma}}\right) = \left(f\left(\boldsymbol{\mu}\right),\,\mathbf{J}_f\left(\boldsymbol{\mu}\right)\boldsymbol{\Sigma}\mathbf{J}_f\left(\boldsymbol{\mu}\right)^{\mathrm{T}}\right), \tag{6}$$

where $\mathbf{J}_f(\boldsymbol{\mu})$ is the Jacobian of $f$ at $\boldsymbol{\mu}$.

**Video-NeRF Optimization.** Given single videos with camera poses calibrated using COLMAP (Schönberger & Frahm, 2016; Schönberger et al., 2016), our video-NeRF model is trained by minimizing the difference between the rendered pixel colors and groundtruth pixel colors. To render pixel colors, we shoot rays and query the scene properties in the scene model and dynamic human model, and re-order all sampled properties based on their distance from the camera center. Then, the pixel color can be calculated through volume rendering (Mildenhall et al., 2021):

$$\hat{\mathbf{C}}\left(\mathbf{r}\right) = \sum_{i=1}^{N} T_i \left(1 - e^{-\sigma_i \delta_i}\right) \mathbf{c}_i, \quad T_i = e^{-\sum_{j=1}^{i-1} \sigma_j \delta_j}. \tag{7}$$

Following HOSNeRF (Liu et al., 2023a), we optimize our video-NeRF representation using by minimizing the photometric MSE loss, patched-based perceptual LPIPS (Zhang et al., 2018) loss, and the regularization losses proposed by Mip-NeRF 360 (Barron et al., 2022) to avoid background collapse, deformation cycle consistency, and indirect optical flow supervisions. Please refer to HOS-NeRF (Liu et al., 2023a) for more details.

## 3.2 Image-based 3D Canonical Space Editing

**Motivation.** Previous video editing works (Khachatryan et al., 2023; Ouyang et al., 2023; Yang et al., 2023; Bar-Tal et al., 2022; Chai et al., 2023) primarily described intended editing through text prompts. However, finer-grained details and the concept's identity are better conveyed through reference images. To this end, we focus on image based editing for finer and direct controllability. As shown in Fig. 2, since our video-NeRF model eases the large-scale motion- and view-change video editing problem into a 3D space editing task, we instead present our image-based 3D editing pipeline to tackle the video editing problem. To better disentangle the foreground and background editings, we propose to edit the foreground canonical space with both the reference subject image and its text description, and edit the background static space with the reference style image.

### 3.2.1 Image-based Foreground 3D Human Space Editing

**Challenges.** Consistent and high-quality image-based video editing requires the edited 3D human space to 1) keep the personalized contents of the reference image; 2) animatable by the human poses from source video; 3) consistent among large-scale view- and motion- changes; and 4) high-resolution. To address these challenges, we design a set of strategies below.

**Reference Image Reconstruction Loss.** We propose to use a reference subject image $\mathbf{I}^{\mathrm{r}}$ to provide finer identity controls and allow for personalized human editing. To ensure that the reference image have a similar human shape with respect to the source human, we leverage ControlNet (Zhang & Agrawala, 2023) to generate the reference subject image conditioned on a source human pose $\mathbf{P}^{\mathrm{r}}$, as exampled in Fig. 2. Then, we use a pretrained monocular depth estimator (Ranftl et al., 2020) to estimate the pseudo depth $\mathbf{D}^{\mathrm{r}}$ of reference subject and use SAM (Kirillov et al., 2023) to obtain its mask $\mathbf{M}^{\mathrm{r}}$. During training, we assume the reference image viewpoint to be the front view (Ref Camera $\mathbf{V}^{\mathrm{r}}$ in Fig. 2) and render the subject image $\hat{\mathbf{I}}^{\mathrm{r}}$ driven by the source human pose $\mathbf{P}^{\mathrm{r}}$ at $\mathbf{V}^{\mathrm{r}}$ under our video-NeRF representation. We additionally compute the rendered mask $\hat{\mathbf{M}}^{\mathrm{r}}$ and depth $\hat{\mathbf{D}}^{\mathrm{r}}$ at $\mathbf{V}^{\mathrm{r}}$ by integrating the volume density and sampled distances along the ray of each pixel. Following Magic123 (Qian et al., 2023), we supervise our framework at $\mathbf{V}^{\mathrm{r}}$ viewpoint using the mean squared error (MSE) loss on the reference image and mask, as well as the normalized negative Pearson correlation on the pseudo depth map.

$$\mathcal{L}_{\mathrm{REC}} = \lambda_{\mathrm{rgb}} \cdot \left\| \mathbf{M} \odot \left( \hat{\mathbf{I}}^{\mathrm{r}} - \mathbf{I}^{\mathrm{r}} \right) \right\|_2^2 + \lambda_{\mathrm{mask}} \cdot \left\| \hat{\mathbf{M}}^{\mathrm{r}} - \mathbf{M}^{\mathrm{r}} \right\|_2^2 \tag{8}$$

$$+ \lambda_{\mathrm{depth}} \cdot \frac{1}{2} \left( 1 - \frac{\mathrm{cov}\left( \mathbf{M}^{\mathrm{r}} \odot \mathbf{D}^{\mathrm{r}}, \mathbf{M}^{\mathrm{r}} \odot \hat{\mathbf{D}}^{\mathrm{r}} \right)}{\sigma\left( \mathbf{M}^{\mathrm{r}} \odot \mathbf{D}^{\mathrm{r}} \right) \cdot \sigma\left( \mathbf{M}^{\mathrm{r}} \odot \hat{\mathbf{D}}^{\mathrm{r}} \right)} \right), \tag{9}$$

where $\lambda_{\mathrm{rgb}}$, $\lambda_{\mathrm{mask}}$, $\lambda_{\mathrm{depth}}$ are the loss weights, $\odot$ is Hadamard product, $\mathrm{cov}(\cdot)$ is the covariance, and $\sigma(\cdot)$ is the standard deviation.

**Score Distillation Sampling (SDS) from 3D Diffusion Prior.** Although the $\mathcal{L}_{\mathrm{REC}}$ can provide supervision on the reference image contents, it only works on the source human pose $\mathbf{P}^{\mathrm{r}}$ at the reference view $\mathbf{V}^{\mathrm{r}}$. To provide more 3D supervision from the reference image, we utilize the Zero-1-to-3 (Liu et al., 2023b) pretrained on the Objaverse-XL (Deitke et al., 2023) as the 3D diffusion prior to distill the inherent 3D geometric and texture information from the reference image using the SDS loss (Poole et al., 2022). Given the 3D diffusion model $\phi$ with the noise prediction network $\epsilon_\phi(\cdot)$, the SDS loss works by directly minimizing the injected noise $\epsilon$ added to the encoded rendered images $\mathbf{I}$ and the predicted noise. Therefore, we render images $\mathbf{I}$ from the 3D human space driven by the source human pose $\mathbf{P}^{\mathrm{r}}$ at random camera viewpoints $\mathbf{V} = [\mathbf{R}, \mathbf{T}]$, and the SDS loss of Zero-1-to-3 (Liu et al., 2023b) can be computed with the reference image $\mathbf{I}^{\mathrm{r}}$ and the camera pose $[\mathbf{R}, \mathbf{T}]$ as condition:

$$\nabla_\theta \mathcal{L}_{\mathrm{SDS}}^{\mathrm{3D}}(\phi, F_\theta) = \lambda_{\mathrm{3D}} \cdot \mathbb{E}_{t,\epsilon} \left[ w(t) \cdot (\epsilon_\phi(\mathbf{z}_t; \mathbf{I}^{\mathrm{r}}, t, \mathbf{R}, \mathbf{T}) - \epsilon) \cdot \frac{\partial \mathbf{I}}{\partial \theta} \right], \tag{10}$$

where $\mathbf{z}_t$ is the noised latent image by injecting a random Gaussian noise of level $t$ to the encoded rendered images $\mathbf{I}$. $w(t)$ is a weighting function that depends on the noise level $t$. $\theta$ is the optimizable parameters of our DynVideo-E.

**SDS from 2D Personalized Diffusion Prior.** The reference image guided supervisions above are limited to edit the 3D human space driven only by the source human pose $\mathbf{P}^{\mathrm{r}}$, and thus are not sufficient to produce a satisfactory 3D human space that can be animated by the frame human poses from source videos. To this end, we further animate the 3D human space with the frame human poses $\mathbf{P}$ from source video and render images $\mathbf{I}$ at random camera poses $\mathbf{V}$, and we further propose to use the 2D text-based diffusion prior (Rombach et al., 2022) to guide these rendered views. However, naively using the 2D diffusion prior hinders the personalization contents learned from the reference image because the 2D diffusion prior tends to imagine the subject contents purely from text descriptions, as valiated in Fig. 4. To solve this problem, we further propose to use 2D personalized diffusion prior that are first finetuned on the reference image using Dreambooth-Lora (Ruiz et al., 2023; Hu et al., 2021). To generate more inputs for Dreambooth-Lora, we augment the reference image with random backgrounds and use Magic123 (Qian et al., 2023) to augment reference image

with multiple views. With such Dreambooth-Lora finetuned 2D personalized diffusion prior, we further employ the 2D SDS loss to guide the editing of the 3D human space animated by frame poses under random camera poses.

$$\nabla_\theta \mathcal{L}_{\text{SDS}}^{\text{2D}}(\phi, F_\theta) = \lambda_{\text{2D}} \cdot \mathbb{E}_{t,\epsilon}\left[\omega(t)\left(\epsilon_\phi(\mathbf{z}_t; y, t) - \epsilon\right) \cdot \frac{\partial \mathbf{I}}{\partial \theta}\right], \quad (11)$$

where $y$ is the text embedding.

**Text-guided Local Parts Super-Resolution.** Due to the GPU memory limitation, our DynVideo-E is trained with $(128 \times 128)$ resolutions, which results in coarse geometry and blurry textures. To tackle this problem, inspired by DreamHuman (Kolotouros et al., 2023), instead of directly increasing training resolutions at the expense of longer training time, we utilize the text-guided local parts super-resolution. Since we rely on the human pose-driven 3D canonical space under "T-pose" configuration, we can accurately render the zoom-in local human body parts by directly changing the camera center. Specifically, we design 7 semantic regions: full body, head, upper body, midsection, lower body, left arm, and right arm, and we accordingly modify the input text prompt and additionally augment these prompts with view-conditional prompts: front view, side view, and back view. It is worth noting that since it is difficult to track the arms position under all human poses, we only zoom-in the arms under the "T-pose" configuration. Such text-guided local parts zoom-ins can large improve the resolution of edited human space and generate much finer geometric and texture details.

### 3.2.2 IMAGED-BASED BACKGROUND EDITING.

We aim at transfering the artistic features of an arbitrary 2D reference style image to our 3D unbounded scene model. As shown in the green flowchart of Fig. 2, we take inspiration from ARF (Zhang et al., 2022) and adopt its nearest neighbor feature matching (NNFM) style loss to transfer the semantic visual details from the 2D reference image $\mathbf{I}^s$ to our 3D unbounded $\Psi^S$. We additionally utilize the deferred back-propagation (Zhang et al., 2022) to directly optimize our model on full-resolution renderings. Specifically, we render the background images $\mathbf{I}$ and extract the VGG (Simonyan & Zisserman, 2014) feature maps $\mathbf{F}$ and $\mathbf{F}^s$ for $\mathbf{I}$ and $\mathbf{I}^s$, respectively. Therefore, $\mathcal{L}_{\text{NNFM}}$ minimizes the cosine distance between the rendered feature map and its nearest neighbor in the reference feature map.

$$\mathcal{L}_{\text{NNFM}} = \lambda_{\text{NNFM}} \cdot \frac{1}{N} \sum_{i,j} \min_{i',j'} D\left(\mathbf{F}(i,j), \mathbf{F}^s(i',j')\right) \quad D(\mathbf{v}_1, \mathbf{v}_2) = 1 - \frac{\mathbf{v}_1^{\text{T}}\mathbf{v}_2}{\sqrt{\mathbf{v}_1^{\text{T}}\mathbf{v}_1 \mathbf{v}_2^{\text{T}}\mathbf{v}_2}}, \quad (12)$$

To prevent the scene model from deviating much from the source contents, we also add an additional l2 loss penalizing the difference between $\mathbf{F}$ and $\mathbf{F}^s$ (Zhang et al., 2022).

### 3.3 TRAINING OBJECTIVES

The training of DynVideo-E consists of 2 stages. Firstly, we reconstruct our video-NeRF model using only source videos as supervision. Secondly, we separately edit the 3D human space and 3D unbounded scene space given reference images and text prompts. After training, we integrate the edited 3D spaces and render our edited videos along source video camera viewpoints.

**Multi-view Multi-pose Training for 3D human space.** As shown in Fig. 2, we design a multi-view multi-pose training process with 3 conditions during training.

- Orange flowchart in Fig. 2: only $\mathcal{L}_{\text{REC}}$ is used to supervise the rendered images under source human pose $\mathbf{P}^r$ at the reference camera view $\mathbf{V}^r$.
- Blue flowchart in Fig. 2: $\mathcal{L}_{\text{SDS}}^{\text{3D}}$ and $\mathcal{L}_{\text{SDS}}^{\text{2D}}$ are jointly used to supervise the rendered images under source human pose $\mathbf{P}^r$ at random camera view $\mathbf{V}$.
- Red flowchart in Fig. 2: only $\mathcal{L}_{\text{SDS}}^{\text{2D}}$ is used to supervise the rendered images under frame human pose $\mathbf{P}$ from source video at random camera view $\mathbf{V}$.

## 4 EXPERIMENTS

**Dataset.** To evaluate our DynVideo-E on both long and short videos, we utilize HOSNeRF (Liu et al., 2023a) dataset with $[300, 400]$ frames per video and NeuMan (Liu et al., 2023a) dataset with

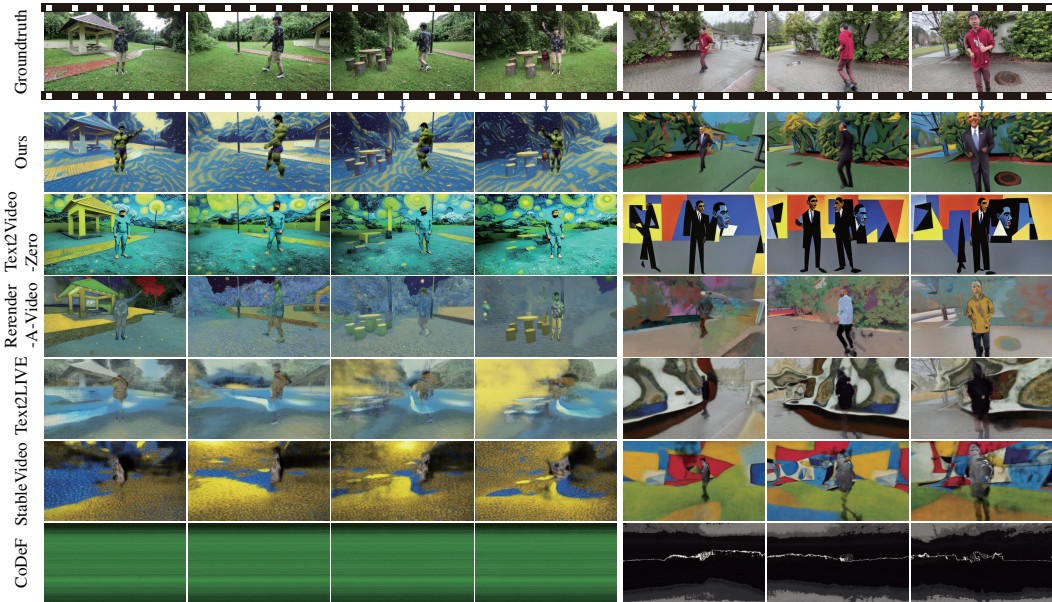

Figure 3: Qualitative comparisons of DynVideo-E and SOTA approaches.

Table 1: Quantitative comparisons of our DynVideo-E against SOTA approaches.

| | METRICS | HUMAN PREFERENCE | | |
|---|---|---|---|---|
| | CLIPScore (↑) | Textual Faithfulness (↑) | Temporal Consistency (↑) | Overall Quality (↑) |
| Text2Video-Zero (Khachatryan et al., 2023) | 26.70 | 9.17 v.s. **90.83 (Ours)** | 21.25 v.s. **78.75 (Ours)** | 12.08 v.s. **87.92 (Ours)** |
| Rerender-A-Video (Yang et al., 2023) | 26.11 | 6.67 v.s. **93.33 (Ours)** | 25.00 v.s. **75.00 (Ours)** | 9.58 v.s. **90.42 (Ours)** |
| Text2LIVE (Bar-Tal et al., 2022) | 22.77 | 3.81 v.s. **96.19 (Ours)** | 26.67 v.s. **73.33 (Ours)** | 9.05 v.s. **90.95 (Ours)** |
| StableVideo (Chai et al., 2023) | 22.02 | 4.29 v.s. **95.71 (Ours)** | 24.29 v.s. **75.71 (Ours)** | 6.19 v.s. **93.81 (Ours)** |
| CoDeF (Ouyang et al., 2023) | 16.77 | 1.25 v.s. **98.75 (Ours)** | 3.75 v.s. **96.25 (Ours)** | 1.25 v.s. **98.75 (Ours)** |
| DynVideo-E (Ours) | **31.31** | - | - | - |

$[30, \, 90]$ frames per video, all at a resolution of $(1280 \times 720)$. In total, we design 24 editing prompts on 11 challenging dynamic human-centric videos to evaluate our DynVideo-E and all baselines.

## 4.1 COMPARISONS WITH SOTA APPROACHES

**Baselines.** We compare our method against five SOTA approaches, including Text2Video-Zero (Khachatryan et al., 2023), Rerender-A-Video (Yang et al., 2023), Text2LIVE (Bar-Tal et al., 2022), StableVideo (Chai et al., 2023), and CoDeF (Ouyang et al., 2023). We utilize Midjourney [1] to generate the text descriptions of the reference images to train these baseline approaches.

**Qualitative Results.** We present a visual comparison of our approach against all baselines in Fig. 3 for a long video (left) and a short video (right). Since both videos contain large motions and viewpoints changes, all baseline approaches fail to edit the foreground or background and cannot preserve a consistent structure. In contrast, our DynVideo-E produces high-quality videos that largely outperform baseline approaches for both long and short videos, and our method consistently edits both the foreground subject and background with correct motions and view changes. We provide more visual comparisons of all methods in Sec. A.1 of appendix. It is worth noting that for challenging videos with large-scale motions and viewpoint changes, CoDeF (Ouyang et al., 2023), Text2LIVE (Bar-Tal et al., 2022), and StableVideo (Chai et al., 2023) largely overfit to input video frames and learn meaningless canonical images or neural atlas, and thus cannot generate meaningful editing results. We show examples of their learned canonical images and neural atlas in Sec. A.1 of appendix.

**Quantitative Results.** We quantify our method against baselines through metrics and human preferences. We measure the textual faithfulness by computing the average CLIPScore (Hessel et al., 2021) between all frames of output videos and corresponding text descriptions. As shown in Tab. 1, our results achieves the highest textual faithfulness score among all SOTA approaches.

---

[1]https://www.midjourney.com/

*Human Preference.* We show the pairwise comparing videos and textual descriptions to raters, and ask them to select their preference videos in terms of textual faithfulness, temporal consistency, and overall quality. We utilize Amazon MTurk [2] to recruit 10 participants for each comparison (Each comparison may recruit different raters), and compute their preferences over all comparisons on all prompts. For each comparison, we show our result and one baseline result (shuffled order in questionnaires), together with textual descriptions to raters and ask their preferences. In total, we collected 1140 comparisons over all pairwise results from 32 different raters. As shown in Tab. 1, we report the comparison "$p_1\%$ v.s. $p_2\%$" where $p_1$ represents the percentage of a baseline is preferred and $p_2$ denotes our method is preferred. As evident in Tab. 1, our method achieves the highest human preference in all aspects and outperforms all baselines by a large margin of $50\% \sim 95\%$.

## 4.2 ABLATION STUDY

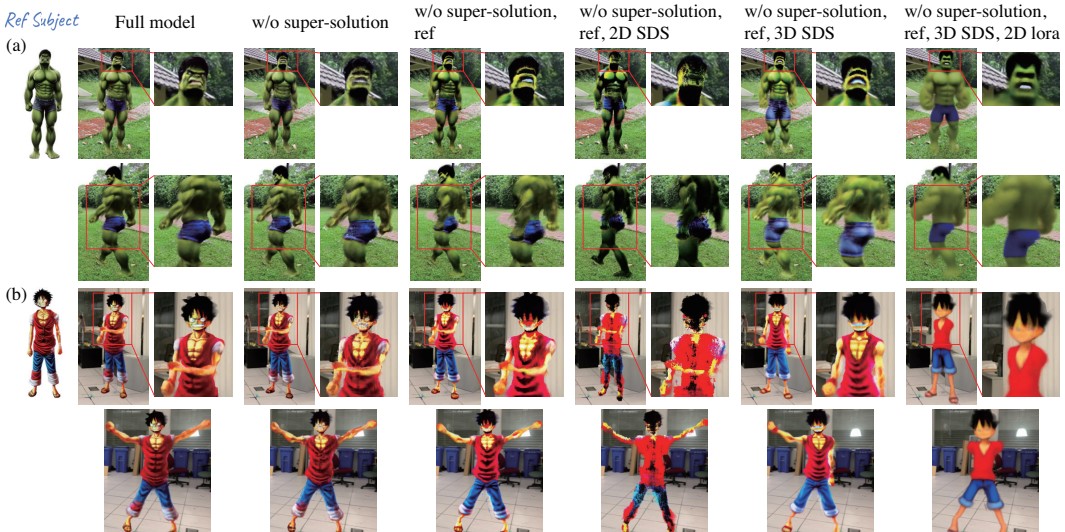

Figure 4: Ablation study on each proposed component for (a) Backpack scene and (b) Lab scene.

We conduct ablation studies on 2 videos from HOSNeRF dataset (Liu et al., 2023a) and NeuMan dataset (Jiang et al., 2022). To evaluate the effectiveness of each proposed component in DynVideo-E, we progressively ablate each component from local parts super-resolution, reconstruction loss, 2D personalized SDS, 3D SDS, and 2D personalization lora. As evident in Fig. 4, the performance of DynVideo-E progressively drops with the disabling of each component, with the full model achieving the best performances, which demonstrates the effectiveness of our designs.

## 5 CONCLUSION

We introduced a novel framework of DynVideo-E to consistently edit large-scale motion- and view-change human-centric videos. To tackle the challenges of such a task, we first introduced dynamic NeRF as our video representation to ease the video editing task into a static 3D editing task with edited contents accurately propagated to the entire video via the deformation fields. Then, we proposed a set of designs to edit the canonical spaces using reference images. These included multi-view multi-pose Score Distillation Sampling (SDS) from both 2D personalized diffusion priors and 3D diffusion priors, reconstruction losses on the reference image, text-guided local parts super-resolution, and style transfer for 3D background space. Finally, extensive experiments demonstrated DynVideo-E produced significant improvements over SOTA approaches.

**Limitations and Future Work.** Although DynVideo-E achieves remarkable results in challenging video editing, its first stage of reconstructing the video-NeRF model is time-consuming. Combining voxel representation or hash grids into the video-NeRF model can largely reduce the training time, and we leave it as a faithful future direction.

---

[2]https://requester.mturk.com/

## 6 REPRODUCIBILITY STATEMENT

To ensure the reproducibility of DynVideo-E: 1) We will release our code, data, and model weights to the community. 2) We use publicly available datasets for experiments and will release the reference images we used in experiments. 3) We provide all comparison videos on the anonymous project page `https://dynvideo-e.github.io`. (4) We provide additional experimental comparisons in Sec. A.1 of the appendix, as well as the human evaluation setups in Sec. A.2 of the appendix.

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

# A APPENDIX

## A.1 ADDITIONAL EXPERIMENTAL RESULTS

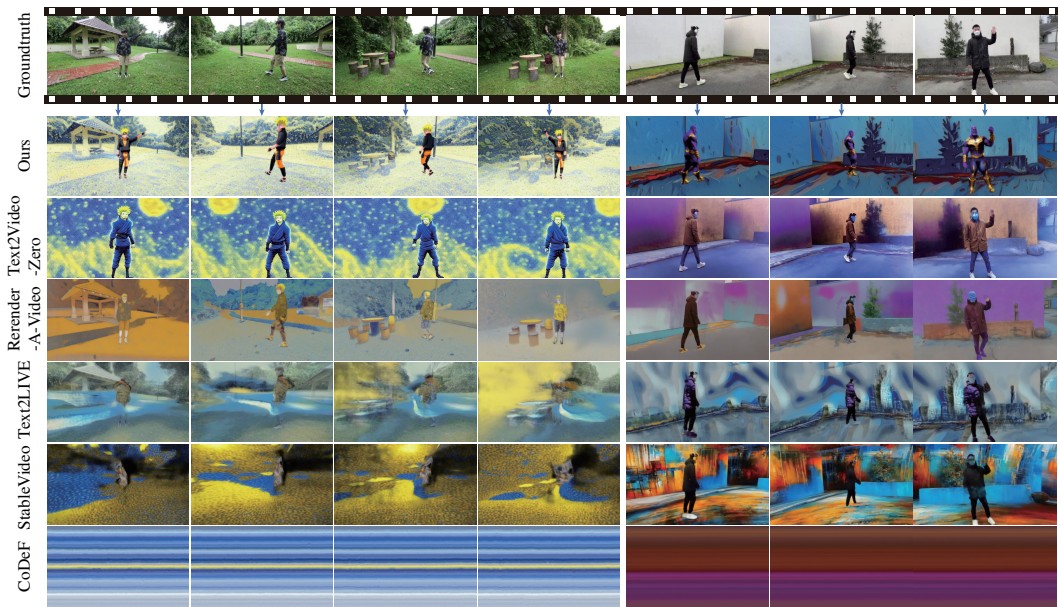

Figure 5: More qualitative comparisons of DynVideo-E against SOTA approaches.

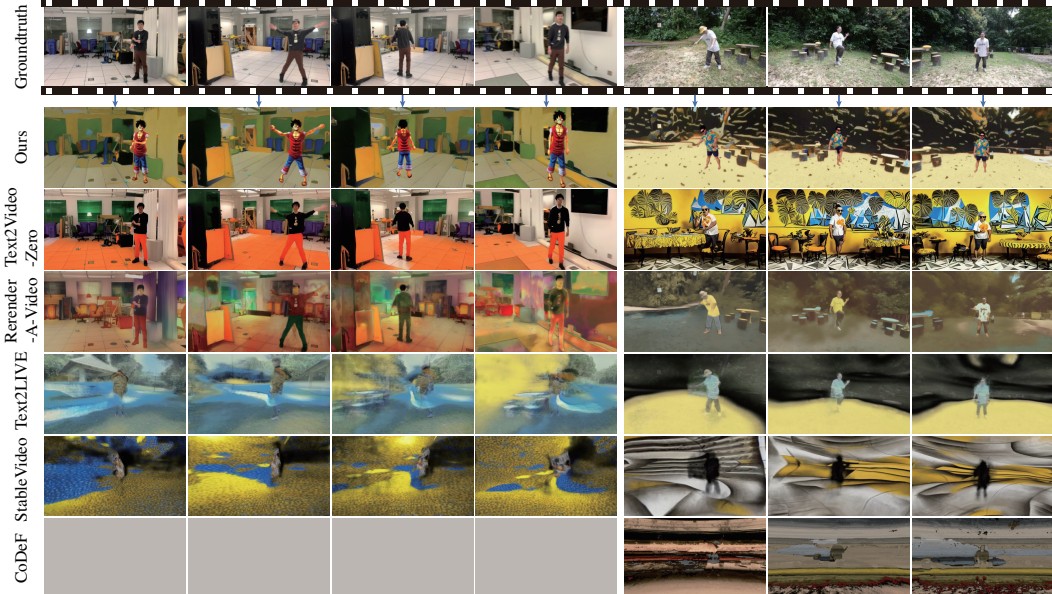

Figure 6: More qualitative comparisons of DynVideo-E against SOTA approaches.

**More Qualitative Results.** We present two more visual comparisons of our approach against all baselines in Fig. 5 and Fig. 6. As shown in the figures, our DynVideo-E achieves the best performances with photorealistic edited videos, which clearly demonstrates the superiority of our model against other approaches on editing large-scale motion- and view- change human-centric videos. Comparing the left (long) and right (short) video of Fig. 5, we find that baseline approaches perform better on short videos than long videos, but still none of them can generate the correct sub-

ject "Thanos" due to the large subject motions and viewpoint changes in videos. In contrast, our DynVideo-E produces high-quality editing results on both short and long videos.

Figure 7: Visualization of canonical images from CoDeF (Ouyang et al., 2023), and foreground and background atlas from Text2LIVE (Bar-Tal et al., 2022) and StableVideo (Chai et al., 2023) on (a) HOSNeRF dataset (Liu et al., 2023a) and (b) NeuMan dataset (Jiang et al., 2022).

**Visualization of Canonical Images from CoDeF (Ouyang et al., 2023) and Atlas from Text2LIVE (Bar-Tal et al., 2022) and StableVideo (Chai et al., 2023).** For challenging videos with large-scale motions and viewpoint changes, CoDeF (Ouyang et al., 2023), Text2LIVE (Bar-Tal et al., 2022), and StableVideo (Chai et al., 2023) largely overfit to input video frames and learn meaningless canonical images or neural atlas, and thus cannot generate meaningful editing results. We show several examples of their learned canonical images (Ouyang et al., 2023) and neural atlas (Bar-Tal et al., 2022; Chai et al., 2023) in Fig. 7, where Text2LIVE (Bar-Tal et al., 2022), and StableVideo (Chai et al., 2023) utilizes the same foreground and background atlas during training. As shown in Fig. 7, canonical images and atlas all fail to represent the challenging large-scale motion- and view-change videos, and thus cannot generate satisfactory editing results. In addition, the atlas performs better for short videos in NeuMan dataset (Jiang et al., 2022) with a better background atlas, but the foreground atlas still cannot represent the humans with large motions. In contrast, our DynVideo-E represents video with the dynamic NeRF to effectively aggregate the large-scale motion- and view- change video information into a 3D background space and a 3D canonical human space, and achieves high-quality video editing results by editing the 3D spaces.

## A.2 EXAMPLE OF HUMAN PREFERENCE QUESTIONNAIRE

We utilize Amazon MTurk [3] to recruit raters to rate our pairwise comparing videos. For each comparison, we show our result and one baseline result (shuffled order in questionnaires), together with textual descriptions to raters and ask their preferences. In total, we collected 1140 comparisons over all pairwise results from 32 different raters. Fig. 8 illustrate one comparison example in our questionnaires.

---

[3]https://requester.mturk.com/

Figure 8: One comparison example from our questionnaires.

