# OpenReview forum: "DynVideo-E: Harnessing Dynamic NeRF for Large-Scale Motion- and View-Change Human-Centric Video Editing"
_ICLR.cc/2024/Conference — ICLR 2024 Conference Withdrawn Submission_

### Official Review · Reviewer_cDNu · 2023-10-30

**Soundness:** 3 good
**Presentation:** 3 good
**Contribution:** 3 good
**Rating:** 5
**Confidence:** 3

**Summary:**

This work present a novel framework of DynVideo-E that for the first time introduces the dynamic NeRF as the video representation for large-scale motion- and view-change human-centric video editing. With a set of customized designs and training strategies, it outperforms SOTA approaches by a large margin on human preference.

**Strengths:**

Long term video editing consistency has been improved by a large margin;
Qualitative video results shows great performance gain.

**Weaknesses:**

The whole customized process is a bit lengthy with a lot of customization, which potentially makes the reproductivity difficult.

**Questions:**

Could you explain a bit more about the part of Text-guided Local Parts Super-Resolution?

---

> ### Author Response · Authors · 2023-11-14
> **Response to Reviewer cDNu (1/2)**
>
> Thank you very much for your review! We have prepared a comprehensive response and hope it satisfactorily addresses your concerns.
>
> **Q1: Reproductivity.**
> > Weakness 1: The whole customized process is a bit lengthy with a lot of customization, which potentially makes the reproductivity difficult.
>
> **A1:** Thanks for your comment but we respectfully disagree with your opinion. Similar to the 2D-video representation based video editing methods, our video-NeRF based video editing pipeline consists of two parts: reconstruction and editing. Regarding the reviewer’s question about customization, our NeRF-video reconstruction module **first represents the video with a dynamic NeRF**, and our editing module further edits the foreground and background given reference images as input. To enable distilling personalized contents from the 2D diffusion prior, we propose to use **2D personalized diffusion prior** that is finetuned on the reference image using Dreambooth-Lora. Other 2D-video representation based methods such as CoDeF [A], StableVideo [B], and Text2LIVE [C] also require first reconstructing the video using 2D representations and further editing such 2D representations with designed editing models. For example, StableVideo [B] first separately reconstructs the foreground and background atlas of a video and separately edits them with editing models, and it additionally uses video depth information to improve consistency. **In comparison with these approaches, we think our method is not lengthy in terms of customization.**
>
> As mentioned in the reproducibility statement of the main paper, to ensure the reproducibility of DynVideo-E: 1) **We will release our code, data, and model weights to the community upon acceptance.** 2) We use **publicly available datasets** for experiments and will release the reference images we used in experiments. 3) We provide **all comparison videos** on the anonymous project page [https://dynvideo-e.github.io](https://dynvideo-e.github.io). (4) We provide **additional experimental comparisons as well as the human evaluation setups in the appendix**. We also **provide all the model designs and details in the main paper**, and the **ablation study** quantitatively (Tab. A in the next response) and qualitatively (Fig. 4 in the main paper) **demonstrate the effectiveness of our designs**. We additionally provide the **24 360° free-viewpoint renderings** for edited foreground and background as well as edited subjects on [https://anonymous.4open.science/r/DynVideo-E-5791/](https://anonymous.4open.science/r/DynVideo-E-5791/).
>
> **Therefore, we believe we can assure reproductivity through the above steps.**
>
> [A] Ouyang, H., Wang, Q., Xiao, Y., Bai, Q., Zhang, J., Zheng, K., ... & Shen, Y. (2023). Codef: Content deformation fields for temporally consistent video processing. arXiv preprint arXiv:2308.07926.
>
> [B] Chai, W., Guo, X., Wang, G., & Lu, Y. (2023). Stablevideo: Text-driven consistency-aware diffusion video editing. In Proceedings of the IEEE/CVF International Conference on Computer Vision (pp. 23040-23050).
>
> [C] Bar-Tal, O., Ofri-Amar, D., Fridman, R., Kasten, Y., & Dekel, T. (2022, October). Text2live: Text-driven layered image and video editing. In European conference on computer vision (pp. 707-723). Cham: Springer Nature Switzerland.

---

> ### Author Response · Authors · 2023-11-14
> **Response to Reviewer cDNu (2/2)**
>
> **Q2: Text-guided Local Parts Super-Resolution.**
>
> > Question 1: Could you explain a bit more about the part of Text-guided Local Parts Super-Resolution?
>
> **Q2:** Thanks for your question. Due to the GPU memory limitation, our DynVideo-E is trained with (128 × 128) low resolutions, resulting in blurry texture and coarse geometry. To refine the details of the human body, we utilize the text-guided local parts super-resolution to **render and supervise the local parts of zoom-in humans**, such as the head, arms, and different parts of bodies. As such, **the local parts can be rendered in 128 × 128, which improves their effective resolution**. To supervise such local parts renderings, we accordingly modify the input text prompt with respect to the rendered parts. For example, if we want to edit the human to the Hulk, and if we render the head of the human, we augment the text with “A hulk head” to distill Stable Diffusion using the 2D SDS loss. In total, we designed 7 semantic regions: full body, head, upper body, midsection, lower body, left arm, and right arm, and we randomly selected these semantic regions by placing the camera in the corresponding positions during training. **We provide 8 visualization examples of text-guided local parts super-resolution sampled during training in** [https://anonymous.4open.science/r/DynVideo-E-5791/Superresolution.pdf](https://anonymous.4open.science/r/DynVideo-E-5791/Superresolution.pdf). As shown in this figure, even though all figures are rendered in 128 × 128 resolutions, rendering local parts can largely improve the effective resolution and thus we can supervise the detailed geometry and textures with diffusion priors.
>
> However, how to correctly locate the camera position to the zoom-in body parts is the key problem. Since we rely on the human pose-driven 3D canonical space under the ”T-pose” configuration, we can accurately render the zoom-in local human body parts by **directly locating the camera close to the corresponding parts**. Since it is difficult to track the arm's position under all human poses due to occlusions, we only zoom in on the arms under the ”T-pose” configuration. In addition, depending on the position of the cameras around the human, we **augment the local parts’ text prompts with view-conditional prompts**: front view, side view, and back view.
>
> To better evaluate the effectiveness of the super-resolution module, we compute the **quantitative results for our ablation studies** by computing the average cosine similarity between the CLIP image embeddings of all frames of output edited videos and the corresponding reference subject image. As shown in the quantitative ablation of Tab. A and the qualitative ablation of Fig. 4 in the main paper, **our text-guided local parts super-resolution can largely improve the resolution of edited human space and generate much finer geometric and texture details**.
>
>
> | CLIP score (↑) | Full      | w/o super-solution | w/o super-solution, ref | w/o super-solution, ref, 2D SDS | w/o super-solution, ref, 3D SDS | w/o super-solution, ref, 3D SDS, 2D lora |
> |------------|-----------|--------------------|-------------------------|---------------------------------|---------------------------------|------------------------------------------|
> | Backpack   | **0.756** | 0.736              | 0.728                   | 0.679                           | 0.711                           | 0.698                                    |
> | Lab        | **0.647** | 0.645              | 0.617                   | 0.517                           | 0.613                           | 0.539                                    |
>
> *Table A. Quantitative results on each proposed component of our method for the Backpack scene and Lab scene.*

---

> ### Author Response · Authors · 2023-11-15
> **Follow-up on Initial Rebuttal Submission**
>
> Dear Reviewer cDNu,
>
> Thank you for your valuable feedback on our submission. We have read your comments carefully and have addressed them in our rebuttal. We would be grateful if you could acknowledge if our responses have addressed your comments. We would also be happy to engage in further discussions if needed. Thank you again for your time and consideration.

---

### Official Review · Reviewer_6Wks · 2023-11-01

**Soundness:** 3 good
**Presentation:** 3 good
**Contribution:** 3 good
**Rating:** 6
**Confidence:** 5

**Summary:**

The paper proposes a dynamic NeRF-based approach to handle video editing in 3D space. The proposed method uses a deformation field to propagate the edits to the entire video. The authors introduce several design improvements that enhance the editing performance. The experimental results demonstrate the effectiveness of the proposed approach, as shown by both qualitative and quantitative evaluations.

**Strengths:**

1. The paper is well-written, easy to follow, and features illustrative figures that effectively convey the concepts.

2. The proposed method is well-motivated and adeptly addresses existing limitations by employing 4D representations for video editing. This approach integrates motion and view changes, while the deformation field guarantees consistency throughout the edited video.

3. The experimental evaluation is comprehensive, offering both qualitative and quantitative evidence that demonstrates the superiority of the proposed method over baseline approaches.

**Weaknesses:**

1. One limitation of the proposed method is the requirement for calibrated camera poses, which may not be readily obtainable for all videos. This constraint could potentially restrict the applicability of the approach in certain scenarios or require additional preprocessing steps to estimate camera poses.

2. Similar to other recent NeRF-based generation and editing methods, the proposed approach can be time-consuming. This factor may hinder its adoption in real-time applications or situations where rapid editing is necessary.

3. As a NeRF-based approach, the edited videos should ideally support free-viewpoint rendering. However, the qualitative results presented in the paper only show editing results with the aligned timestamp and viewpoint as the input video. This aspect raises questions about the method's ability to generate consistent and accurate results across different viewpoints, which is a key advantage of NeRF-based approaches.

**Questions:**

1. COLMAP can sometimes fail with moving objects in the scene for camera calibration. It would be helpful if the authors could provide more details on how to run COLMAP in such cases and if there are any specific parameters or settings that can be adjusted to improve its performance.

2. It would be beneficial if the authors could provide an analysis of the time required for a single editing operation using the proposed approach. This information would help to understand the practical feasibility of the proposed method for real-world applications.

3. It would be interesting to see an edited result with an arbitrary camera trajectory or a static human subject while the camera is moving (e.g., bullet time effect) instead of only showing results with viewpoints aligned to the input video. This would highlight the versatility of the proposed method and its potential for a wide range of applications.

4. As another dynamic NeRF-based approach for video editing, the paper does not mention the work by Zhang et al., "Editable free-viewpoint video using a layered neural representation" (SIGGRAPH 2021). It would be helpful if the authors could provide a brief comparison between their proposed method and the approach presented in Zhang et al.'s work in the related works section.

---

> ### Author Response · Authors · 2023-11-14
> **Response to Reviewer 6Wks (1/2)**
>
> Thank you very much for recognizing our work and the valuable comments! We have prepared a comprehensive response and hope it satisfactorily addresses your concerns.
>
> **Q1: About camera pose.**
> > Weakness 1: The requirement for calibrated camera poses could potentially restrict the applicability of the approach in certain scenarios or require additional preprocessing steps to estimate camera poses.
>
> > Question 1: COLMAP can sometimes fail with moving objects in the scene for camera calibration. How to run COLMAP in such cases and if there are any specific parameters or settings that can be adjusted to improve its performance.
>
> **A1:** Thank you for your helpful advice. To harness the dynamic NeRF as the video-NeRF representation, we need to obtain the camera poses of human-centric videos as a priori information using COLMAP. However, directly using COLMAP to calibrate dynamic videos may sometimes fail with moving objects. To solve this problem, we **estimate the masks for the moving objects**. Because COLMAP supports calibrating dynamic videos with moving object masks as input, we can mask out the dynamic moving areas and only calibrate the videos with static areas. This can largely **improve the camera pose calibration accuracy** for dynamic videos.
>
> Since we focus on human-centric videos, we use a pre-trained segmentation algorithm such as Mask-RCNN [A] to directly estimate the masks of moving humans. Such algorithms are very robust and accurate in segmenting humans, so this pre-process is **very convenient and fast to deploy** and does not require human corrections. In addition, the segmentation masks also help our video-NeRF model separately represent the dynamic human and static background with a dynamic human NeRF model and a static background NeRF model.
>
> Therefore, we believe it is convenient and fast to estimate the human masks of human-centric videos, and the COLMAP can accurately calibrate the camera poses of human-centric videos by masking out the dynamic human regions. We think **this pre-process step is very convenient and automatic, and can handle most human-centric videos**.
>
> [A] He, K., Gkioxari, G., Dollár, P., & Girshick, R. (2017). Mask r-cnn. In Proceedings of the IEEE international conference on computer vision (pp. 2961-2969).
>
> **Q2: About training time.**
> > Weakness 2: Similar to other recent NeRF-based generation and editing methods, the proposed approach can be time-consuming.
>
> > Question 2: Analysis of the time required for a single editing operation using the proposed approach.
>
> **A2:** Thank you for your helpful advice. Editing long-term human-centric videos with large-scale motions and viewpoint changes is a particularly challenging task. We are **the first to tackle this problem** by introducing the **novel 4D representation** to degrade video editing to a 3D space editing problem. It is true that the NeRF-based generation and editing methods are time-consuming, but there have been many excellent works on improving the training speed of NeRF by utilizing the voxel [G] and hash grids [H] representations. Therefore, combining **voxel representation or hash grids** into our DynVideo-E can largely reduce the training time, and we leave it as a faithful future direction.
>
> Specifically, we compare the editing operation time of our method against other approaches on a long video of the HOSNeRF dataset ([300, 400] frames) using a single A100 GPU in Tab. A. Although other approaches are faster than ours, 2D-video representation-based methods such as CoDeF [B], StableVideo [E], and Text2LIVE [F] cannot accurately reconstruct large-scale motion- and view-change videos and thus fail to generate meaningful editing results, as validated in Fig. 7 of the main paper. Text2Video-Zero [C] and Rerender-A-Video [D] fail to edit the challenging human-centric videos with large-scale motion and viewpoint changes and their editing results are highly inconsistent, as shown in Tab. 1 and Fig. 3 of the main paper. Therefore, **previous approaches cannot handle the challenging human-centric videos no matter how many computation resources are provided**. In contrast, **our method is the first work to achieve highly consistent long-term video editing** that outperforms previous approaches by a large margin of 50% ∼ 95% in terms of human preference, and **we leave accelerating our model with voxel or hash grid representation as a faithful future direction.**
>
> | Method | CoDeF [B] | Text2Video-Zero [C] | Rerender-A-Video [D] | StableVideo [E] | Text2LIVE [F] | Ours      |
> |--------|-----------|---------------------|----------------------|-----------------|---------------|-----------|
> | Time   | ~1mins    | 15 mins             | 1.2 hours            | ~1 mins         | ~2 hours      | 7.3 hours |
>
> *Table A. Editing operation time comparison of our method against other approaches.*

---

> ### Author Response · Authors · 2023-11-14
> **Response to Reviewer 6Wks (2/2)**
>
> **Q3: Free-viewpoint rendering.**
> > Weakness 3: Whether the method can generate consistent and accurate results across different viewpoints.
>
> > Question 3: It would be interesting to see edited results with an arbitrary camera trajectory or a static human subject while the camera is moving.
>
> **A3:** Thank you for your helpful advice! **Yes, our method supports free-viewpoint rendering of the edited scenes for both the edited human and background.** We have rendered **360° free-viewpoint renderings (bullet time effect)** for both the edited human and edited background for all the edited scenes of HOSNeRF dataset, which includes 12 challenging bullet time effect videos. We have also rendered the 360° free-viewpoint edited human subjects for all the edited scenes of HOSNeRF and NeuMan dataset, resulting in 12 360° free-viewpoint edited subjects videos. We have uploaded these **24 360° free-viewpoint videos** in [https://anonymous.4open.science/r/DynVideo-E-5791/](https://anonymous.4open.science/r/DynVideo-E-5791/) for your kind reference.
>
> As shown in these rendered videos, our method achieves **very high-quality 360° free-viewpoint renderings of the edited scenes**, while other methods cannot achieve novel view synthesis. We totally agree with you that **this would highlight the versatility of the proposed method and its potential for a wide range of applications**, and we will add this free-viewpoint rendering ability to our model’s highlights and we will include these 360° free-viewpoint renderings in our final paper.
>
>
>
> **Q4: Discussion with Zhang et al. [I].**
> > Question 4: Comparison with Zhang et al. [I]
>
> **A4:** Thanks for your helpful advice. Zhang et al. [I] is one of the pioneering works on editable free-viewpoint video and it presents a spatiotemporal coherent neural layered radiance representation called ST-NeRF. The differences between Zhang et al. [I] and our work are two-fold.
>
> 1.  **Zhang et al. [I] require 16 cameras** to reconstruct a dynamic scene, while **our method requires only 1 camera** to capture in-the-wild videos, which is much more convenient and promising for real-world applications. Both methods can achieve large viewpoint synthesis, but our method can achieve high-quality 360° free-viewpoint renderings of both the edited human and background, while Zhang et al. [I] mainly capture the 180° dynamic scenes.
>
> 2.  Zhang et al. [I] can only achieve simple editing by manipulating the NeRF layer, such as affine transform or duplication as well as temporal editing like retiming performers’ movements. These operations **do not edit the contents of layered NeRF** but only change the ray sampling manner. Therefore, it cannot directly edit the identity or texture/geometry of the human subject and it cannot edit the background. In contrast, our method proposes the image-based 3D space editing pipeline with a set of effective designs based on the proposed multi-view multi-pose score distillation sampling (SDS). With these designs, **we can largely edit the subject identity given only one single reference image**, and we also can edit the style of the background with one reference style image. Therefore, **our method’s editing capability is much more advanced than Zhang et al. [I]**.
>
> We will include the comparison with Zhang et al. [I] in our related works section.
>
> [B] Ouyang, H., Wang, Q., Xiao, Y., Bai, Q., Zhang, J., Zheng, K., ... & Shen, Y. (2023). Codef: Content deformation fields for temporally consistent video processing. arXiv preprint arXiv:2308.07926.
>
> [C] Khachatryan, L., Movsisyan, A., Tadevosyan, V., Henschel, R., Wang, Z., Navasardyan, S., & Shi, H. (2023). Text2video-zero: Text-to-image diffusion models are zero-shot video generators. arXiv preprint arXiv:2303.13439..
>
> [D] Yang, S., Zhou, Y., Liu, Z., & Loy, C. C. (2023). Rerender A Video: Zero-Shot Text-Guided Video-to-Video Translation. arXiv preprint arXiv:2306.07954.
>
> [E] Chai, W., Guo, X., Wang, G., & Lu, Y. (2023). Stablevideo: Text-driven consistency-aware diffusion video editing. In Proceedings of the IEEE/CVF International Conference on Computer Vision (pp. 23040-23050).
>
> [F] Bar-Tal, O., Ofri-Amar, D., Fridman, R., Kasten, Y., & Dekel, T. (2022, October). Text2live: Text-driven layered image and video editing. In European conference on computer vision (pp. 707-723). Cham: Springer Nature Switzerland.
>
> [G] Sun, C., Sun, M., & Chen, H. T. (2022). Direct voxel grid optimization: Super-fast convergence for radiance fields reconstruction. In Proceedings of the IEEE/CVF Conference on Computer Vision and Pattern Recognition (pp. 5459-5469).
>
> [H] Müller, T., Evans, A., Schied, C., & Keller, A. (2022). Instant neural graphics primitives with a multiresolution hash encoding. ACM Transactions on Graphics (ToG), 41(4), 1-15.
>
> [I] Zhang, J., Liu, X., Ye, X., Zhao, F., Zhang, Y., Wu, M., ... & Yu, J. (2021). Editable free-viewpoint video using a layered neural representation. ACM Transactions on Graphics (TOG), 40(4), 1-18.

---

> ### Author Response · Authors · 2023-11-15
> **Follow-up on Initial Rebuttal Submission**
>
> Dear Reviewer 6Wks,
>
> Thank you for your valuable feedback on our submission. We have read your comments carefully and have addressed them in our rebuttal. We would be grateful if you could acknowledge if our responses have addressed your comments. We would also be happy to engage in further discussions if needed. Thank you again for your time and consideration.

---

### Official Review · Reviewer_6oK3 · 2023-11-06

**Soundness:** 2 fair
**Presentation:** 2 fair
**Contribution:** 2 fair
**Rating:** 5
**Confidence:** 4

**Summary:**

This paper presents a method for consistent editing of large-scale motion- and view-change human-centric videos. Specifically, the proposed method exploits dynamic NeRF for the video representation, and integrates several techniques including the Score Distillation Sampling (SDS) from both 2D personalized diffusion priors and 3D diffusion priors, reconstruction losses on the reference image, text-guided local parts superesolution, and style transfer for 3D background space. The experiments validate the effectiveness of the proposed method.

**Strengths:**

+ The paper is clear and easy to follow.
+ Multiple existing techniques are combined into the whole pipeline.

**Weaknesses:**

- About the novelty: The paper is a system paper that combines several existing works, including HOSNeRF, Zero-1-to-3, and Magic123, without too much novel insight. For example, the basic video representation follows the existing work HOSNeRF, and the only difference is the removal of object state designs for the specific task in the paper. Both 3D and 2D priors follow the existing works, Zero-1-to-3, and Magic123. From these points, the novelty of the paper mainly lies in the integration of such existing works.
- About the application scenario: The proposed method relies on the dynamic human NeRF reconstruction, making it limited to human-centric video and less interesting.
- About the experiment: A quantitative comparison is also expected for the ablation. Moreover, only a rather small dataset is utilized for the test, and more videos in the wild are expected.

**Questions:**

Please refer to the weakness part.

---

> ### Author Response · Authors · 2023-11-14
> **Response to Reviewer 6oK3 (1/3)**
>
> Thank you very much for your review! We have prepared a comprehensive response and hope it satisfactorily addresses your concerns.
>
> **Q1: About the novelty.**
> > Weakness 1: The paper is a system paper that combines several existing works, including HOSNeRF, Zero-1-to-3, and Magic123, without too much novel insight.
>
> **A1:** Thanks for your comment but we respectfully disagree with your opinion. The novelty of this paper lies in three-fold:
>
> Firstly, we **for the first time introduce the dynamic NeRF as the innovative video representation** for large-scale motion- and view-change human-centric video editing. This is a clear gap for existing frame-wise editing methods and 2D-video representation based editing methods. As supported by reviewer 6Wks, “The proposed method is **well-motivated and adeptly addresses existing limitations** by employing **4D representations** for video editing.”
>
> Secondly, we propose **a set of effective designs and training strategies for the image-based dynamic human editing and background space editing** in our video-NeRF model. As mentioned in the main paper, our distillation modules do take inspiration from Zero-1-to-3 and Magic123, but our designs are different from theirs in the following aspects: 1) Zero-1-to-3 and Magic123 are designed for static single object generation, while our method focuses on **editing the dynamic human body**. To achieve that, we designed a **multi-view multi-pose distilling strategy** to edit the 3D dynamic human under **various dynamic animation human poses and camera poses**. 2) To enable distilling personalized contents from the 2D diffusion prior, we propose to use the **2D personalized diffusion prior** that is finetuned on the reference image using Dreambooth-Lora. 3) We include other designs such as **text-guided local parts super-resolution** to improve the effective resolution of edited human space and **style transfer module** to edit the 3D background space. Our complete model finally achieves editing both the dynamic foreground human body and static unbounded background.
>
> Thirdly, DynVideo-E not only supports editing challenging human-centric videos with large-scale motion and viewpoint changes, it can also achieve novel view synthesis and even **360° free-viewpoint renderings** for edited human and background, as shown in the **24 free-viewpoint rendering videos** in [https://anonymous.4open.science/r/DynVideo-E-5791/](https://anonymous.4open.science/r/DynVideo-E-5791/). Extensive experiments show that DynVideo-E significantly outperforms SOTA approaches on two challenging datasets **by a large margin of 50% ~ 95%** in terms of human preference.

---

> ### Author Response · Authors · 2023-11-14
> **Response to Reviewer 6oK3 (2/3)**
>
> **Q2: About the application scenario.**
> > Weakness 2: The proposed method relies on the dynamic human NeRF reconstruction, making it limited to human-centric video and less interesting.
>
> **A2:** Thanks for your comment but we respectfully disagree with your opinion. As stated in the title, our DynVideo-E focuses on large-scale motion- and view-change human-centric video editing. **Human-centric videos are very important and common in real-world applications** such as **movie creation, social media, and virtual reality**, and they are **very challenging in terms of large-scale motions and viewpoint changes**. Therefore, there have been **many excellent research works on human modeling** (such as NeuralBody [A], HumanNeRF [B], NeuMan [C], and HOSNeRF [D]) and **human editing** (such as Control4D [E], AvatarStudio [F], etc). **Many companies also focus on human-centric video editing and generation** such as Wonder Studio ([https://wonderdynamics.com/](https://wonderdynamics.com/)), Synthesia ([https://www.synthesia.io/homepage](https://www.synthesia.io/homepage)), etc. Therefore, **we believe human-centric video editing is a very interesting and challenging task** that **has drawn much interest from both the research community and industry fields**, and our DynVideo-E adeptly addresses existing limitations by employing 4D representations for video editing and significantly improves the performance in this task. **Editing more general dynamic scenes is also very interesting and we leave it as a faithful future direction.**
>
> [A] Peng, S., Zhang, Y., Xu, Y., Wang, Q., Shuai, Q., Bao, H., & Zhou, X. (2021). Neural body: Implicit neural representations with structured latent codes for novel view synthesis of dynamic humans. In Proceedings of the IEEE/CVF Conference on Computer Vision and Pattern Recognition (pp. 9054-9063).
>
> [B] Weng, C. Y., Curless, B., Srinivasan, P. P., Barron, J. T., & Kemelmacher-Shlizerman, I. (2022). Humannerf: Free-viewpoint rendering of moving people from monocular video. In Proceedings of the IEEE/CVF conference on computer vision and pattern Recognition (pp. 16210-16220).
>
> [C] Jiang, W., Yi, K. M., Samei, G., Tuzel, O., & Ranjan, A. (2022, October). Neuman: Neural human radiance field from a single video. In European Conference on Computer Vision (pp. 402-418). Cham: Springer Nature Switzerland.
>
> [D] Liu, J. W., Cao, Y. P., Yang, T., Xu, Z., Keppo, J., Shan, Y., ... & Shou, M. Z. (2023). HOSNeRF: Dynamic Human-Object-Scene Neural Radiance Fields from a Single Video. In Proceedings of the IEEE/CVF International Conference on Computer Vision (pp. 18483-18494).
>
> [E] Shao, R., Sun, J., Peng, C., Zheng, Z., Zhou, B., Zhang, H., & Liu, Y. (2023). Control4D: Dynamic Portrait Editing by Learning 4D GAN from 2D Diffusion-based Editor. arXiv preprint arXiv:2305.20082.
>
> [F] Pan, M. M., Elgharib, M., Teotia, K., Tewari, A., Golyanik, V., Kortylewski, A., & Theobalt, C. (2023). AvatarStudio: Text-driven Editing of 3D Dynamic Human Head Avatars. arXiv preprint arXiv:2306.00547.

---

> ### Author Response · Authors · 2023-11-14
> **Response to Reviewer 6oK3 (3/3)**
>
> **Q3: About the experiment.**
> > Weakness 3: A quantitative comparison is also expected for the ablation. Moreover, only a rather small dataset is utilized for the test, and more videos in the wild are expected.
>
> **A3:** Thanks for your helpful advice on the quantitative comparison of the ablation study. To provide the quantitative results of our ablation study, we compute the CLIP image embeddings on all frames of output edited videos and the corresponding reference subject image, and we report **the average cosine similarity between the CLIP image embeddings of all frames of output edited videos and the corresponding reference subject image**. We report the quantitative results for the 2 ablation videos of the main paper (Fig. 4 of main paper) in Tab. A, where we progressively ablate each component from local parts super-resolution, reconstruction loss, 2D personalized SDS, 3D SDS, and 2D personalization lora. As shown in Tab. A, **the CLIP score progressively drops with the disabling of each component, with the full model achieving the best performances, which clearly demonstrates the effectiveness of our designs.**
>
> | CLIP score (↑) | Full      | w/o super-solution | w/o super-solution, ref | w/o super-solution, ref, 2D SDS | w/o super-solution, ref, 3D SDS | w/o super-solution, ref, 3D SDS, 2D lora |
> |------------|-----------|--------------------|-------------------------|---------------------------------|---------------------------------|------------------------------------------|
> | Backpack   | **0.756** | 0.736              | 0.728                   | 0.679                           | 0.711                           | 0.698                                    |
> | Lab        | **0.647** | 0.645              | 0.617                   | 0.517                           | 0.613                           | 0.539                                    |
>
> *Table A. Quantitative results on each proposed component of our method for the Backpack scene and Lab scene.*
>
> We respectfully disagree with your opinion on the scale of the experimentation dataset. We extensively evaluate our DynVideo-E and all baselines on both long videos (HOSNeRF dataset) and short videos (NeuMan dataset), resulting in **24 different editing prompts on 11 challenging in-the-wild dynamic human-centric videos**. We **compare the number of scenes/videos with other NeRF-based scene editing methods and non-NeRF-based video editing methods in Tab. B.** As validated in Tab. B, our experimental datasets are **extensive and adequate** compared to the NeRF-based editing methods and non-NeRF-based video editing methods. In addition, our evaluated dataset is **much more challenging** than other approaches in terms of **video length, large-scale motions, and large-scale viewpoint changes**. Therefore, we believe **our experiment's datasets are extensive, adequate, and challenging** to evaluate the proposed method and baselines.
>
> | Method               | DreamEditor [G] | NeuralEditor [H] | Rerender-A-Video [I] | Control4D [J] | StableVideo [K] | Text2LIVE [L] | Ours                    |
> |----------------------|-----------------|------------------|----------------------|---------------|-----------------|---------------|-------------------------|
> | No. of scenes/videos | 6               | 8                | 15                   | 9             | 6               | 7             | **24 prompts on 11 videos** |
>
> *Table B. Comparison of the number of experimentation scenes/videos between our method and other approaches.*
>
> [G] Zhuang, J., Wang, C., Liu, L., Lin, L., & Li, G. (2023). DreamEditor: Text-Driven 3D Scene Editing with Neural Fields. arXiv preprint arXiv:2306.13455.
>
> [H] Chen, J. K., Lyu, J., & Wang, Y. X. (2023). NeuralEditor: Editing Neural Radiance Fields via Manipulating Point Clouds. In Proceedings of the IEEE/CVF Conference on Computer Vision and Pattern Recognition (pp. 12439-12448).
>
> [I] Yang, S., Zhou, Y., Liu, Z., & Loy, C. C. (2023). Rerender A Video: Zero-Shot Text-Guided Video-to-Video Translation. arXiv preprint arXiv:2306.07954.
>
> [J] Shao, R., Sun, J., Peng, C., Zheng, Z., Zhou, B., Zhang, H., & Liu, Y. (2023). Control4D: Dynamic Portrait Editing by Learning 4D GAN from 2D Diffusion-based Editor. arXiv preprint arXiv:2305.20082.
>
> [K] Chai, W., Guo, X., Wang, G., & Lu, Y. (2023). Stablevideo: Text-driven consistency-aware diffusion video editing. In Proceedings of the IEEE/CVF International Conference on Computer Vision (pp. 23040-23050).
>
> [L] Bar-Tal, O., Ofri-Amar, D., Fridman, R., Kasten, Y., & Dekel, T. (2022, October). Text2live: Text-driven layered image and video editing. In European conference on computer vision (pp. 707-723). Cham: Springer Nature Switzerland.

---

> ### Author Response · Authors · 2023-11-15
> **Follow-up on Initial Rebuttal Submission**
>
> Dear Reviewer 6oK3,
>
> Thank you for your valuable feedback on our submission. We have read your comments carefully and have addressed them in our rebuttal. We would be grateful if you could acknowledge if our responses have addressed your comments. We would also be happy to engage in further discussions if needed. Thank you again for your time and consideration.

---

### Author Response · Authors · 2023-11-14
**Reply to AC and all reviewers**

We thank all the reviewers for their insightful and valuable comments. We also appreciate that the core contributions of our paper and the quality of our results are recognized in the review:

1.  The proposed method is **well-motivated** and **adeptly addresses existing limitations** by employing **4D representations** for video editing. (Reviewer 6Wks)

2.  **Long term video editing consistency** has been **improved by a large margin**; Qualitative video results show **great performance gain**. (Reviewer cDNu)

3.  The paper is **clear and easy to follow**. (Reviewer 6oK3)

Below, we respond to the individual concerns of each reviewer, and we are very happy to discuss if you have any further questions.

---

### Author Response · Authors · 2023-11-17
**Follow-up on Initial Rebuttal Submission**

Dear Reviewers,

Thank you for your valuable feedback on our submission. We have read your comments carefully and have addressed them in our rebuttal. We would be grateful if you could acknowledge if our responses have addressed your comments. We would also be happy to engage in further discussions if needed. Thank you again for your time and consideration.

Best regards,

Authors of paper 5791